# Fetal Heart Rate Monitoring Implemented by Dynamic Adaptation of Transmission Power of a Flexible Ultrasound Transducer Array

**DOI:** 10.3390/s19051195

**Published:** 2019-03-08

**Authors:** Paul Hamelmann, Massimo Mischi, Alexander F. Kolen, Judith O. E. H. van Laar, Rik Vullings, Jan W. M. Bergmans

**Affiliations:** 1Department of Electrical Engineering, Eindhoven University of Technology, 5612 AP Eindhoven, The Netherlands; m.mischi@tue.nl (M.M.); r.vullings@tue.nl (R.V.); j.w.m.bergmans@tue.nl (J.W.M.B.); 2Philips Research, 565 AE Eindhoven, The Netherlands; alex.kolen@philips.com; 3Máxima Medical Center, 5504 DB Veldhoven, The Netherlands; judith.van.laar@mmc.nl

**Keywords:** fetal heart rate monitoring, Doppler Ultrasound, dynamic apodization, wearable sensors

## Abstract

Fetal heart rate (fHR) monitoring using Doppler Ultrasound (US) is a standard method to assess fetal health before and during labor. Typically, an US transducer is positioned on the maternal abdomen and directed towards the fetal heart. Due to fetal movement or displacement of the transducer, the relative fetal heart location (fHL) with respect to the US transducer can change, leading to frequent periods of signal loss. Consequently, frequent repositioning of the US transducer is required, which is a cumbersome task affecting clinical workflow. In this research, a new flexible US transducer array is proposed which allows for measuring the fHR independently of the fHL. In addition, a method for dynamic adaptation of the transmission power of this array is introduced with the aim of reducing the total acoustic dose transmitted to the fetus and the associated power consumption, which is an important requirement for application in an ambulatory setting. The method is evaluated using an in-vitro setup of a beating chicken heart. We demonstrate that the signal quality of the Doppler signal acquired with the proposed method is comparable to that of a standard, clinical US transducer. At the same time, our transducer array is able to measure the fHR for varying fHL while only using 50% of the total transmission power of standard, clinical US transducers.

## 1. Introduction

After the introduction of the first commercial fetal monitor in 1968, Cardiotocography (CTG), a simultaneous registration of fetal heart rate (fHR) and uterine contraction (UC), is now routinely applied in clinical practice to assess the fetal well-being in the antepartum and intrapartum period [1]. The objective of CTG is to minimize the risk of fetal morbidity and mortality, to determine optimal timing of delivery, and to identify fetuses at risk [2,3,4]. While CTG has a very high sensitivity, its specificity in the prediction of fetal compromise is poor, which has led to a significant increase in unnecessary caesarian sections [5]. Currently, due to the high inter and intra-observer variability in the interpretation of the CTG recordings [6], computerized approaches for CTG analysis are under development [6,7,8]. In particular, the analysis of features obtained from the fHR recording and analysis of fHR variability are gaining more and more interest, as they may directly reflect the functioning of fetal autonomous regulation [9,10,11,12,13]. 

The fundamental requirement for enabling computerized analysis is accurate and robust recording of the fHR. Up to date, the most common technology to measure the fHR is Doppler Ultrasound (US). An US transducer, operating in a pulsed-wave Doppler mode, is positioned on the maternal abdomen and directed towards the location of the fetal heart. From the received Doppler signal, fetal-heart periodicity is determined using algorithms which typically make use of an autocorrelation function (ACF) to obtain an estimate of the fHR [14,15]. However, the quality of the Doppler signal strongly depends on the correct position of the US transducer relative to the fetal heart location (fHL). More recently, non-invasive recordings of the fetal electrocardiographic signal (ECG) using abdominal surface electrodes are under investigation [16,17,18]. This promising technique, which may use different types and numbers of electrodes located concentrically around the maternal abdomen [16], is less dependent on sensor position and is suitable for measurements on mothers with a high body-mass-index (BMI) [19], and possibly provides insight into morphological changes in the ECG signal. However, signal processing of the electrophysiological signal remains challenging due to low signal-to-noise ratio (SNR), for instance because of the superimposed maternal ECG, and is not yet routinely used in clinical practice [16,17]. 

So as to obtain an US Doppler signal with good quality, the transducer needs to be placed by a skilled clinician. However, positioning can be challenging in preterm pregnancies due to the small size of the fetal heart and because the fetus can move freely within the uterus [20]. Once the clinician has found a good US signal, she/he fixates the transducer on the maternal abdomen using a flexible belt to enable continuous recordings of the fHR at a specific fetal heart location. However, due to changing fetal-heart location or due to displacement of the US transducer on the maternal abdomen, fHR recordings often show severe episodes of signal loss making them unsuitable for clinical interpretation [21,22,23]. As a result, clinicians often have to come to the bedside for manual repositioning of the US transducer, which is time consuming and an undesired disruption of the clinical workflow. In addition, relevant information about fetal condition can be missed. This can result in a delay in intervention and may put the fetus at risk. Recently, to improve the work flow in obstetrical care we proposed a method which facilitates improved US transducer positioning by estimating the fHL relative to the transducer [20].

In this research, we propose to measure the fHR using a flexible transducer array, which can be applied to an arbitrary curvature of the maternal abdomen and doesn’t need to be repositioned at all. The basic idea is to increase the measurement volume by using multiple US transducers in such a way that the fHR can be measured at all possible fetal heart-locations. This is implemented using an algorithm which dynamically adapts the active transmitting elements, with the objective of minimizing total transmission power, since this is a major clinical concern even if safety guidelines are fulfilled [24], and maximizing signal-to-noise (SNR) ratio. The optimal algorithm settings are investigated for different fHLs as well as movement velocities of the fetal heart with respect to the US transducer using a dedicated in-vitro beating fetal-heart setup.

## 2. Materials and Methods

### 2.1. Design of a Flexible Ultrasound Transducer Array 

Commercially available US transducers used for fetal heart rate monitoring typically consist of multiple US transducer elements. These elements are arranged on a rigid grid in a hexagonal or circular pattern and driven simultaneously by a parallel electrical interconnection of the elements. This effectively forms an array aperture, which insonifies a cone-like measurement volume of several centimeters in diameter. When the measurement volume is increased by adding multiple transducer elements to the array aperture, they cannot be placed on such a rigid grid due to the curvature of the abdomen. Instead they would need to be integrated into a flexible material. In that way, the array can be wrapped around the maternal abdomen. We used the following production process to create a prototype of such a flexible transducer array (Figure 1a). 

As a first step, a mold is created using Polymethylmethacrylat (PMMA) plates. The available piezoelectric transducer elements (PZTs), operating at center frequency *f*_0_ = 1 MHz, have diameter *d* = 10 mm and thickness *t* = 2 mm. Accordingly, 10-mm wide holes with element spacing of *d_k_* = 10 mm are drilled into a 1-mm thick PMMA plate following a hexagonal configuration (Figure 1b). Subsequently, the piezoelectric transducer elements are connected to coaxial cables and pressed into the holes such that they are fixated in the specified configuration, preventing them from moving or tilting during the next production steps. In the third step, liquid Polydimethylsiloxan (PDMS, Dow Corning Sylgard 184, mixed at ratio of 10:1 of base to catalyst) is poured into the mold to cover the PZTs. PDMS is a silicone which is flexible, robust, and has similar acoustic properties as human tissue; therefore, it is used in many biomedical US applications [25,26]. Mixing occurs in a shaker/centrifuge to remove air bubbles. To remove bubbles introduced after pouring, the mold is placed in a low vacuum (0.8 bar) for 5 min and abruptly vented, which forces bubbles to rise and escape at the PDMS surface. For curing the PDMS, the mold is placed for 3 h in an oven at *T* = 90 °C. Note that the bottom half of the transducer elements is now embedded into PDMS. This allows for flipping the array and removing the thin PMMA plate without displacing the transducer elements. Subsequently, a second layer of PDMS is poured into the mold onto the front side of the previous cast to embed the transducer elements completely. The thickness of the PDMS layer above the transducer surface is approximately 1 mm. Again, bubbles are removed by a low vacuum treatment and the PDMS is cured at *T* = 90 °C, prior to removal from the mold. Pouring the second PDMS layer in a later step has the advantage that a regular and air-free layer above the surface of the transducer elements is created. 

In this prototype, in total *M* = 37 transducer elements are used, distributed over a rectangular surface of 8 cm by 15 cm (see Figure 1b). This configuration is large enough to allow evaluating the feasibility and performance of the algorithm proposed in the subsequent sections. In clinical practice, a larger array or different shape may be required to cover all potential heart locations.

It should be noted that no backing material is used, which would prevent excessive vibrations of the transducer elements. However, as the system operates in a pulsed-wave Doppler mode using long pulse lengths, this omission causes a negligible error during measurements. Driving of the transducer elements is done via an US research platform (Vantage 256, Verasonics, Inc., Kirkland, WA, USA), which allows full control in receive and transmit mode.

### 2.2. Ultrasound Beam Profile

In [20], the transmitted US beam profile of the same transducer elements used in the flexible array was characterized using a needle hydrophone (Precision Acoustics Ltd., Dorchester, UK). Here, we explicitly distinguished the US beam profile during transmit and receive mode. 

It was found that the transmitted US beam of a single piezo element can be approximated in the far field by a narrow diverging cone, which has a linearly increasing width for increasing depths, which is in good agreement with the well-known Fraunhofer-approximation of the US beam profile [27]. Due to reciprocity, the same characterization applies to the US beam in receive mode. Using experiments and simulations, it was shown that transmission with multiple elements, positioned in a rigid gird, leads to an US beam profile with multiple side lobes and interferences [20,28]. It should be noted that a group of seven concentrically arranged transducer elements (e.g., elements with index *i* = 1, *i* = 3, *i* = 4, *i* = 6, *i* = 8 and *i* = 9 in Figure 1b), corresponds to the array aperture of a commercially available US transducer [20,28]. In receive mode, all elements are active and read out individually. Due to the arbitrary curvature of the flexible transducer array, the exact locations of the transducer elements are not known. This makes it impossible to approximate the US beam of the new developed transducer array [27]. 

### 2.3. Doppler Processing

For the acquisition of the Doppler signals, the transducer elements are driven with pulse repetition frequency PRF = 1.6 kHz. The driving pulse has center frequency *f*_0_ = 1 MHz and pulse duration of 20 cycles. The peak-to-peak voltage applied to the transducer elements is *V_pp_* = 1.6 V and can be controlled by setting a transmit apodization function *a_i_*, where the index *i* indicates the element number (defined as in Figure 1b). Apodization functions are well-known window functions in US beamforming, where they are commonly applied to reduce the presence of undesired side lobes [29]. The apodization *a_i_* can have values in the range of 0 ≤ *a_i_* ≤ 1, where *a_i_* = 0 completely switches off the element and *a_i_* = 1 sets the voltage of the transmitted pulse to *V_pp_*. 

To obtain a Doppler signal with each transducer element, for each transmitted US pulse, the received raw Radio-Frequency (RF) signals are bandpass filtered around *f*_0_ and digitized using a sampling frequency of *f_s_* = 4 MHz. Subsequently, the signals are demodulated using a time gate which corresponds to a sample volume depth of SVD = 6 cm to 16 cm, since this is the range in which the fetal heart is typically located during fHR measurements. In that way, each transmitted US pulse leads to one sample of the Doppler signals in each element. In order to remove the contribution of static and slow moving objects, the Doppler signal is filtered using a clutter filter with cut-off frequency *f_c_* = 50 Hz [30]. 

Subsequently, for each transducer element, the power ***P***
*=* (*P*_1_, *P*_2_, …, *P_M_*) of the Doppler signal is determined using the mean squared value in a sliding time window *w*. It is important to choose the *w* large enough to guarantee that at least one heartbeat is present within the signal, such that ***P*** allows for identifying which elements are most suitable to measure the fHR. In this study, the window size is set to *w* = 2 s. The received Doppler power is used to dynamically adjust the apodization function *a_i_*, i.e., to control the transmit power of the elements. This will be further explained in Section 2.5.

### 2.4. Heart Rate Estimation

Methods for estimating the fHR from Doppler US signals typically make use of an autocorrelation function, where the envelope of the Doppler signal is correlated with a delayed copy of itself [15,31]. This allows finding periodic patterns, i.e., the fetal heartbeat, in a signal obscured by noise. 

The envelope signal is determined by estimating the magnitude of the Doppler signal using the Hilbert transform [31] and the implemented ACF is defined as:(1)$$\mathrm{ACF}\left[\tau \right]=\sum_{j=0}^{N-\tau }x\left[n+j\right]x\left[n+j+\tau \right],\;0\leq \tau \leq N$$where *x* is the analyzed signal, *n* the first sample in the autocorrelation window, *N* the length of the ACF window *w*_acf_ expressed in samples and τ the delay [31]. Further, the ACF is normalized by its value ACF[0]. By finding the delay at which the normalized ACF has a maximum, the periodicity of the Doppler signal can be determined, giving an estimate for the fHR. The search for the maximum in the normalized ACF is limited to the range of τ corresponding to 60 bpm to 180 bpm. The fHR estimation is considered as too low quality when the peak in the normalized ACF is below an empirically determined threshold of 0.6, as suggested in [31].

As mentioned before, in commercially available systems the transducer elements are electronically interconnected, such that only one combined Doppler signal is received. Consequently, the ACF is computed for that combined Doppler signal. Since we are measuring the Doppler signals in all transducer elements individually, with a total number of M = 37 elements, we can also estimate the fHR from the individual Doppler signals, by following a similar approach as proposed by Voicu et al. [15]. To reduce the total amount of data, which has to be processed, the Doppler signals are down sampled to 400 Hz, before the envelope is detected. Every time the fHR estimation is executed, the ACF is computed for the last *w*_acf_ = 2 s (i.e., *N* = 800 samples) of the envelope signals. The length *w*_acf_ is important as it determines how many beats are within the time window. Having multiple beats within the time window increases the robustness of the estimation while reducing its accuracy, since small variations in heart rate are averaged out. Setting *w*_acf_ = 2 s guarantees that at very low fetal heart rates of 60 beats per minute (bpm), at least two heart beats are within the ACF window.

In Figure 2, an example of a Doppler signal, measured on an *in-vitro* beating fetal heart setup, which will be explained in Section 2.6.2., and the corresponding ACF can be seen.

### 2.5. Dynamic Adaptation of Transmit Apodization

An important consideration for dynamic adaptation of transmit apodization (DAA) is that transducer elements not directed toward the fetal heart should not be used for transmission, as they (1) will only increase acoustic dose, (2) increase power consumption and (3) will not be able to induce a measured signal related to the fHR. DAA can be achieved by modulating the peak-to-peak voltage *V_pp_* of the driving pulse applied to the individual transducer elements using the apodization function *a_i_*. By dynamically updating *a_i_* based on the fetal heart location, the power consumption and acoustic dose can be minimized and limited during measurements, while being able to continuously obtain an fHR recording. A block diagram of the DAA process is presented in Figure 3. An initial apodization function is used to drive the transducer elements. The received raw RF data is subsequently processed to obtain the Doppler signals of each individual channel, as described in Section 2.3.

Ideally, the apodization should be updated as soon as a heartbeat has occurred, since this will give new information on whether different elements receive higher Doppler power, as a consequence of changing fetal position, and whether the apodization has to be adjusted accordingly. However, in practice the exact timing of the beat is not known. Therefore, it is chosen to execute the DAA with an apodization update frequency *f_a_*. A technical limitation of the adopted research system is that updating the apodization function does not occur instantaneously. When a new apodization function is set, the next transmit-receive sequence is delayed, causing an undesired gap in the acquired Doppler signals. This limitation of the system is circumvented by updating the apodization function only at *f_a_* = 0.5 Hz, such that 2 s of continuous Doppler Data can be recorded to feed into the ACF algorithm. It should be noted that the apodization update frequency *f_a_*, in fact, determines how fast the DAA method can adapt to new fHLs, and hence, defines the maximum velocity with which the fetal heart is allowed to move within the abdomen without completely leaving the US beam. At a theoretically possible minimal fHR = 60 bpm, one heart beat every second contributes to the measured Doppler power, and hence, the apodization update frequency should be set at least to *f_a_* = 1 Hz. Due to the technical limitations of the system, setting this desired *f_a_* is not possible and hence the maximum velocity which can be tracked is reduced. However, if the method is implemented on a dedicated system, a higher *f_a_* could be considered, allowing the DAA to adapt to faster fHL velocities, if clinically required.

In a clinical setting, the fetal heart changes its location when the fetus moves. This movement can be either a fast change of fetal presentation or a slower body movement. It is commonly known that fetuses with lower gestational age change their location more rapidly and frequently than at a later stage of pregnancy, where the freedom of motion is restricted by the limited space in the uterus [32].

The input for the apodization update is the Doppler power and the fHR estimate of each individual channel. The strength of the Doppler power gives an indication on which elements are directed towards the fHR. However, Doppler artefacts, for example caused by limb movement, may also contribute to the measured Doppler power. In order to prevent that the DAA sets *a_i_* according to these Doppler artefacts, it is always checked whether the signal obtained in the specific transducer element is able to measure an fHR of sufficient quality, as described in Section 2.4. The updated apodization function is then used for the next transmit-receive sequence.

The total reduction in transmission power *P*_total_ is dependent on how exactly the apodization function is defined. We hypothesize that an apodization function tailored to the size and location of the fetal heart will allow acquiring Doppler signals with similar SNR, defined as the average SNR of all Doppler signals acquired with the individual transducer elements (see also Section 2.6.2), compared to the situation where all elements are transmitting. In addition, we hypothesize that when the fetal heart location changes with a specific velocity, it is beneficial to use a broader US beam, such that the DAA method can adapt to the new fetal heart locations.

To test these hypotheses, we considered two different strategies for updating the apodization function based on the received Doppler power, viz. a single element apodization and a window-based apodization. It should be noted that for both strategies, in this research, the apodization is only applied in transmit mode. In receive mode, no apodization is applied. The working principle of the different DAA methods is illustrated in Figure 4 and discussed in detail in the following subsections.

#### 2.5.1. Single Element Apodization

When the fetal heart lies directly within the US beam of a specific transducer element, it is likely that this element will receive the strongest Doppler power. As soon as the fetal heart moves within the measurement range of another transducer element, eventually that element will receive the highest Doppler power. A straightforward solution to update *a_i_* dynamically is to use only that element for transmission, which receives the highest Doppler power. Accordingly, the apodization function can be defined as
(2)$$a_{i}=\{\begin{array}{c} 1,\;|\;P_{i}=\underset{i}{\arg\;\max\;P}\\ 0,\;|\;P_{i}\neq \underset{i}{\arg\;\max\;P} \end{array}$$
This apodization adaptation is based on the assumption that the fetal heart only slowly changes its location, i.e., it is still within the US beam of the single active transducer element when the DAA is executed. This DAA method has the advantage of using a minimal transmission power. In the remainder of this article, this method will be referred to as DAA1.

#### 2.5.2. Window-Based Apodization

It is unlikely that only a single element is directed towards the fetal heart. Therefore, SNR might improve when multiple transducer elements are used for transmission. Furthermore, the speed with which the fetal heart location is allowed to change can be higher using multiple elements compared to the single transmit DAA1. To distribute the transmission power among the transducer elements a window function can be used which defines the transmit apodization *a_i_* set to the individual transducer elements.

The center of this window function is determined as the power center *C* = {*C_x_, C_y_*} of the received Doppler power:(3)$$C_{x}=\frac{\sum_{i=1}^{M^{\prime }}P_{i}X_{i}}{\sum_{i=1}^{M^{\prime }}P_{i}};\;C_{y}=\frac{\sum_{i=1}^{M^{\prime }}P_{i}Y_{i}}{\sum_{i=1}^{M^{\prime }}P_{i}},$$where *X_i_* and *Y_i_* are the coordinates of the individual transducer elements, when the transducer array is in a complete flat position, and *M*′ are the elements which provide an fHR with sufficient quality (see Section 2.4.). This prevents that the apodization is erroneously set because of elements which receive a high power only due to motion artefacts. It should be noted that by using Equation (3), the measured Doppler power in the individual transducer elements of an arbitrary curved array is projected onto the transducer elements of a flat array. In that way, the power center *C* can be determined without the need of measuring the individual transducer positions. The power center *C* provides information on where the signals receiving the highest Doppler power are located within the transducer array. However, in the case of a curved array it cannot directly be interpreted as the location of the fetal heart. In a flexible transducer array the element positions are unknown, and therefore, it is challenging to predict what the received Doppler power will be, given a specific fHL. Once *C* is determined, a window function can be used to set a smooth apodization function *a_i_*. We consider two different window functions and evaluate their influence on transmission power and SNR.

First, a Gaussian window based apodization function is defined as:(4)$$a_{i}=\exp\left(-\left(\frac{{\left(X_{i}-C_{x}\right)}^{2}}{\sigma^{2}}+\frac{{\left(Y_{i}-C_{y}\right)}^{2}}{\sigma^{2}}\right)\right),$$where *σ* defines the width of the Gaussian window. This method allows for smoothly distributing the transmission power among the transducer elements. In the remainder of this article, this method will be referred to as DAA2.

In the third method (DAA3), a group of adjacent elements is completely switched on, i.e., their apodization *a_i_* is set to 1. For the elements surrounding this group, the apodization exponentially decays with rate λ. The aim is to make full use of all the transducer elements which are directed towards the fetal heart. The apodization function is defined as:(5)$$a_{i}=\{\begin{array}{c} 1,\;d\leq D\\ e^{-\lambda \left(d-D\right)},\;d>D \end{array}$$where *d* is the Euclidean distance between the power center *C* and the element position and the parameter *D* defines the window width in which the apodization of the transducer elements is set to one. Note that setting the decay constant λ to large values effectively causes that only a group of adjacent elements is fully transmitting and all other elements are turned off.

In Figure 5a,b, the total transmit power *P*_total_ as function of the parameter settings is illustrated. Since *P*_total_ is dependent on the location of the power center *C*, the graphs show *P*_total_ for the situation when *C* coincides (*C*_on_) with an element position and when *C* lies in between elements (*C*_off_), describing the minimal and maximal transmission power with the set parameters. Figure 5c,d show the applied apodization values for selected DAA parameters, which are used in the experiments described in Section 2.6. It should be noted that while having similar *P*_total_, the DAA3 method with large λ distributes the power among a lower number of elements compared to the DAA2 method. This means that using the DAA3 method, locally a stronger US wave will be transmitted toward the fetal heart.

#### 2.5.3. (Re)-Initialization

When heart rate recordings are started, the initial apodization is set to *a*_0_ = (1, …, 1), i.e., all transducer elements are simultaneously transmitting an US wave and the measured Doppler power can subsequently be used to set the first apodization values according to (2), (4) and (5), respectively. With active DAA, whenever not a single valid fHR can be determined, e.g., due to strong motion artifacts caused by UCs or when the array is temporarily removed from the maternal abdomen, the apodization is reset to these initial values. By re-initializing *a_i_*, the most suitable transmit elements are selected by the respective DAA methods as soon as a transducer element is able to estimate the fHR with sufficient quality.

### 2.6. Experiments

#### 2.6.1. Experimental Setup

For testing the flexible transducer array and the DAA-methods, an in-vitro beating-heart setup was realized. The reader is referred to [20], where this setup is explained in detail. Here, the setup (see Figure 6a) is described briefly. To mimic a beating heart of a fetus, a chicken heart is threaded on fishing strings, which are attached to the wall of a water tank. Another fishing string pulls at the chicken heart and brings it into a beat-like motion pattern along the *z*-direction. It should be noted that in real recordings, the shape of the envelope signals will be different due to multiple events during a heartbeat, e.g., the contraction of the ventricles and closing and opening of the valves. However, the DAA methods do not depend on the exact waveform and frequency content of the signal, as they are based on the power of the Doppler signals solely. The flexible array is mounted on a translation stage and the mount allows bending the flexible array to simulate the situation when the array is positioned on the maternal abdomen. Further, by translating the array, a changing fetal heart location within the uterus can be mimicked. For more realistic measurements, an acoustic absorber is positioned in between the array and the chicken heart.

#### 2.6.2. Experimental Design

Two types of measurements, *static* measurements and *dynamic* measurements (see Figure 6b), were conducted to evaluate the performance of the three different DAA methods and to validate that the flexible transducer array can be used to measure the fHR irrespective of the heart location.

In the *static* experiments, the fetal heart was centrally located at a fixed position, i.e., in front of element *i* = 19 (see Figure 6b), at depth *z* = 10 cm, which is a typical distance of the fetal heart with respect to the abdominal surface. The fHR was set to 120 bpm. The different DAA methods were applied and after a short settling period, a 10 s long Doppler measurement was recorded. Besides the application of the DAA methods, it was also investigated how the received Doppler power is distributed among the transducer elements when transmitting with all elements simultaneously (TX_all_) or with a selected single element.

In the *dynamic* experiments, the fHL with respect to the transducer array changed by translation of the transducer array. In this study, it was chosen to limit the direction in which the fetal heart changes its location to one direction (*x*-direction) to test if the DAA can successfully adapt to the new heart locations. Therefore, the heart was moved from the left to the right side of the array at depth *z* = 10 cm at constant velocity. It was ensured that at the beginning and end of the translation, the heart was still within the field of view of the curved array. In that way, the quality of the Doppler signals obtained with a flat and with a curved array could be compared without being affected by translation out of the measurement range. Furthermore, as described before, updating the apodization function using the US research system causes a delay in the acquisition. Therefore, while the apodization is being set the motor of the translation stage was stopped to avoid that the heart is displaced to a position, which did not correspond to the most recent apodization function. We considered three different speeds, *v_m_* = 2 mm/s, *v_m_* = 5 mm/s, *v_s_* = 10 mm/s to mimic a fetal heart location change with slow, moderate, and fast speed, respectively. All experiments were performed for both a flat and a curved array. The curvature of a curved array is defined by the radius r (see Figure 6b). During the experiments with a curved array, the radius was set to *r* = 34 cm, which corresponds to the approximate curvature of the maternal abdomen at term. Note that with this curvature the focus is deeper than the typically required measurement depth of *z* = 5 cm to *z* = 20 cm. When the array is placed on the abdomen of a mother with a higher BMI, the array is less curved and the focus is at an even larger depth.

Due to strong phase delays (see inset of Figure 7), we decided to determine the SNR of each individual Doppler signal instead of combining all signals into one Doppler signal from which a single SNR value could be obtained. We assessed the SNR as:(6)$$\mathrm{SNR}=10\log_{10}\left(\frac{P_{A}}{P_{P}}\right),$$where *P_A_* and *P_P_* are the Doppler power measured in the active and passive phase of each Doppler signal, respectively (see Figure 7). It should be noted that the two events visible in the Doppler signal during the active phase correspond to forward and backward motion of the heart relative to the transducer. Subsequently, the average of all SNR values was calculated ($\bar{\mathrm{SNR}}$) to obtain a single outcome measure for comparison.

As the achieved $\bar{\mathrm{SNR}}$ should be evaluated with respect to the transmission power, which is needed to achieve that specific $\bar{\mathrm{SNR}}$, a cost function was defined as:(7)$$\zeta =P_{total}\left[dB\right]-\bar{\mathrm{SNR}}\left[dB\right].$$

A low-cost function ζ indicates that a high $\bar{\mathrm{SNR}}$ is measured or only little power is required to obtain this specific $\bar{\mathrm{SNR}}.$

## 3. Results

In Figure 8, the Doppler power received in the individual transducer elements of the array is visualized to show the influence of a fixed array curvature (*r* = 34 cm) on the measured Doppler power compared to a flat array. During these experiments, the heart was centrally located in front of the transducer array and either all elements were fully transmitting (Figure 8a,b), or only element *i* = 9 was transmitting (Figure 8c,d). When all elements of the flat array are transmitting (Figure 8a), a group of seven adjacent elements in the center of the array receives the highest Doppler power. Because the US beams are diverging and overlapping at larger depths, it appears that the area covered by the elements receiving high Doppler power is larger than the size of the chicken heart. When the array is bent, more elements at the side are directed towards the fetal heart. It should be noted that during the experiments the array was bent along the *x*-direction. This can also be seen in Figure 8b, where elements along that direction receive higher Doppler power, while in the *y*-direction the received Doppler power remains similar.

The effect of the curvature can also been seen for the experiments where only element *i* = 9 was active during transmission. For a flat array, the transmitted US wave of element *i* = 9 does not insonify the chicken heart and hence only little Doppler power is measured. When the curvature is bent, the US beam of that element is shifted towards the heart location. Interestingly, from Figure 8d it can be seen that the element transmitting the US wave is not the element which receives the highest power.

For all experiments, the received Doppler power is not completely symmetric and smoothly distributed. This is because the heart, and the movement direction of the heart during one beat, is not completely symmetric. In addition, for the situation where all elements are transmitting, there may be strong local variations in US pressure, due to constructive and destructive interference of the US waves.

In Figure 9, the measured $\bar{\mathrm{SNR}}$ values for all experiments are shown for different DAA methods. For all experiments, it can be seen that the highest $\bar{\mathrm{SNR}}$ is obtained using all elements simultaneously (TX_all_). However, as one can see from the graphs, one can achieve a similar performance in terms of $\bar{\mathrm{SNR}}$ while using only a group of elements, determined by the settings of the DAA method. This is logical since a large part of the transmitted US wave is not insonifying the chicken heart and is not reflected back to the transducer elements. When using a custom subset of transducer elements (for example using DAA2, *D* = 17.7 and λ = 0.2 in Figure 9a), a locally similar US wave is generated. Not surprisingly, the lowest $\bar{\mathrm{SNR}}$ is achieved when only a single-element apodization (DAA1) is used or when the settings of the window-based methods are such that only a single element is transmitting (i.e., DAA2: *σ* = 10 mm; DAA3: λ = 10 mm^−1^, *D* = 10 mm; compare Figure 5c). When the width of the window of DAA2 and DAA3 is increasing, the $\bar{\mathrm{SNR}}$ increases as well, since a larger part of the chicken heart is insonified. Having an infinitely large window width would effectively set the respective DAA methods to the TX_all_-method. For DAA3, a very slow decay λ has the same effect.

This general trend of improved $\bar{\mathrm{SNR}}$ for increasing window width holds also for the dynamic experiments, where the transducer array is translated through the water tank at different velocities. As expected, the overall signal quality is reduced, indicated by the reduced $\bar{\mathrm{SNR}}$_,_ with increasing velocities. Interestingly, it can be seen that for velocities *v_m_* = 5 mm/s to *v_m_* = 10 mm/s, the TX_all_ methods and DAA methods with large window width show an $\bar{\mathrm{SNR}}$ reduction by approximately 5 dB, while the DAA1 and DAA methods with small window width show an $\bar{\mathrm{SNR}}$ reduction by approximately 10 dB. This is because at higher velocities the heart moves too fast out of the US beam created by a single element. In that case, the signal quality is reduced before the DAA method has updated the apodization according to the new heart location.

It is striking that the $\bar{\mathrm{SNR}}$ values measured in the static experiments are higher for a curved array, compared to those measured with a flat array (Figure 9a), but consistently lower in the dynamic experiments (Figure 9b–d).

In Figure 10, the cost function (Equation (7)) is shown for all performed experiments using the different apodization methods. The cost function takes into account how much transmission power is needed to obtain a specific $\bar{\mathrm{SNR}}$. A DAA method using a single transmitting element, or a small number of elements with lower apodization, has a low total transmission power *P*_total_, resulting in a relatively low cost. On the other hand, using higher transmission power leads to improved $\bar{\mathrm{SNR}}$_._ This tradeoff is in particular observable for the DAA3 method with different window width *D*. A small window width *D* leads to a low $\bar{\mathrm{SNR}}$ while for larger window width *D* the required transmission power is too high.

Figure 11 shows an example of how the apodization adapts to a changing fetal heart location. In the top panel, the apodization values of each transducer element are shown over time, and in the lower panel, the respective Doppler power measured for each element is depicted. In the beginning, the heart is located on one side of the array and after 6 s, the DAA method is turned on. Then, after 10 s, the heart is translated through the measurement volume of the transducer array. It is clearly visible that the DAA-method is able to adapt to the new fetal heart location. It is also visible that the apodization function is updated according to the preceding power values and that within the 2 s before the next apodization update is executed, the fHL has changed already considerably. This is noticeable by the fact that the apodization appears to lag behind the measured Doppler power, suggesting that a faster apodization update frequency may be needed. At approximately 25 s, the heart leaves the field of view of the transducer array, and the method reinitializes using the initial apodization function. When the heart re-enters the measurement range, the DAA automatically adapts its apodization to track the fetal heart.

## 4. Discussion

The results from the experiments presented in Section 3, show that the flexible transducer array allows for measuring the fHR independently of the fetal heart location and that the DAA method is capable of selecting the transmitting elements which are directed towards the fetal heart. The performance of the method and the gain in terms of reduced power consumption is clearly dependent on movement velocity of the fetal heart, curvature of the flexible transducer array, as well as the parameter settings of the respective DAA-method.

### 4.1. Effect of the DAA Parameter Settings

Throughout all experiments, it can be seen that the DAA3-method with width D = 17.7 mm outperforms the other methods in terms of the lowest costs (Figure 10). Interestingly, when λ = 10 mm^−1^, the DAA3-method works better for the static experiment and the experiment with low velocity *v_m_* = 2 mm/s, while setting λ = 0.2 mm^−1^ has a positive effect on the performance for the experiments at higher velocities. This can be explained by the fact that having a slightly wider transmitted beam is beneficial to avoid that the heart is moving out of the measurement volume too quickly.

The lower cost of the DAA3-method compared to the DAA2-method supports the hypothesis that it is beneficial to consolidate the total transmission power over a smaller number of elements, which are directed towards the heart. For the DAA2-method with large *σ*, and the DAA3-method with slow decay parameter λ, many elements far away from the power center are transmitting with low apodization value. The weak US waves generated by those elements do not reach the target, and hence, do not contribute to improved $\bar{\mathrm{SNR}}$.

### 4.2. Effect of the Array Curvature

When using a curved instead of a flat transducer array, more elements are directed toward the location of the heart in the experimental setup. Irrespectively of the exact shape of the transmitted US beam, more elements will receive US waves reflected from the heart, hence increasing the $\bar{\mathrm{SNR}}$ for the curved array relative to a flat array (see Figure 9). Interestingly, for all dynamic experiments the $\bar{\mathrm{SNR}}$ is reduced compared to the measurements using a flat array. The effect of having multiple elements directed toward the heart is also present when the array is translated through the water tank. However, the beneficial effect of having multiple receiving elements is outweighed by the increased interference of US beams for a curved array and the fact that for the DAA methods it is more challenging to determine the best transmitting element (DAA1) or correct location of the power center C (DAA2 and DAA3). For a curved array, it may happen that the element(s) receiving the highest Doppler power do not correspond to those elements which were transmitting. In other words, using a curved array specular reflection will have a higher effect on the received Doppler power compared to a flat array. When the DAA is executed, the apodization function is updated according to the power received in the elements. If the heart moves, the selected apodization function uses transmitting elements which are further away from the optimal transmitting elements than in the case of a flat array. An additional explanation for the observed effect is that by bending the array, the apparent velocity of the heart moving in the water tank is effectively increased. When the heart moves by a small distance at depth *z* = 10 cm, this distance projected on the curved array is larger compared to a flat array.

Using the single-element apodization (DAA1), the overall low power consumption leads to a relatively low cost function (see Figure 10). However, during the experiments it was found that in some special cases the strategy of selecting the transmitting-element based on the maximal received power might results in an instable system. To illustrate this, Figure 12 highlights an example of a measurement where the chicken heart is centrally located in front of a curved transducer array. For visualization purposes, only the received Doppler signals of element *i* = 19 and element *i* = 14 are displayed. It can be seen that at each apodization update, i.e., every 2 s, the apodization switches from element *i* = 19 to element *i* = 14 and vice versa. This is because the angle of incidence is not the same as the angle of reflection, and therefore, always the other element than the transmitting element receives the highest Doppler power, also visible in the amplitude of the Doppler signals in the bottom panel of Figure 9. This effect will also occur when using a window-based apodization with multiple transmitting elements. However, due to the insonification from multiple angles the location of the power center will not drastically change. It can also be observed that the amplitude and shape of the received Doppler signals changes when another single element is used for transmission. This will affect the performance of the fHR estimation, as the shape of two consecutive heartbeats will differ due to the changing insonification angle, resulting in an fHR estimation error using the ACF. This will especially be important if one chooses a shorter time window in the ACF function, which is needed for a beat-to-beat accurate estimation of fHR [31]. The true inter-beat interval may not be correctly determined, as the signal shape changes due to different insonifcation angles rather than different way of how the heart is contracting. Therefore, the strategy to first compute the power center and then define the apodization using a window-based approach may be preferred.

### 4.3. Total Transmission Power in Relation to Clinical US Transducer

The DAA3-method with settings *D* = 17.7 mm and λ = 10 mm^−1^ respectively λ = 0.2 mm^−1^ lead to the lowest cost function (compare Figure 10). Compared to a seven-element US transducer which uses all elements for transmission, these parameters lead to a reduction of total transmission power by −3.7 dB and −3.2 dB. This supports the clinically desired as-low-as-reasonably-achievable principle, which states that acoustic dose should be kept as low as possible, even if exposure limits are below US safety guidelines [24,33]. It should be noted that in our experiments, acoustic attenuation was mimicked by adding an acoustic absorber between the chicken heart and the transducer array and by setting the peak-to-peak voltage to a low value of *V_pp_* = 1.6 V. To measure Doppler signals in a clinical setting where even more acoustic attenuation may be present, higher peak-to-peak voltages may be applied, which increases the total transmission power.

Looking at the experimental results in a static condition (Figure 9a), it can be seen that the $\bar{\mathrm{SNR}}$ using the DAA3-method (*D* = 30 mm; λ = 10 mm^−1^) is comparable to the maximal $\bar{\mathrm{SNR}}$. This method uses in fact a group of six or seven adjacent elements with apodization function of one (compare also Figure 5c). This supports the design choice of having seven transducer-elements within some commercially available transducers applied for fHR monitoring. Using more elements to create a larger measurement volume would not improve the $\bar{\mathrm{SNR}}$ further, albeit at the expense of increasing radiative load.

### 4.4. Recommendation

In this work, the received power determines the apodization of the transmitting elements, while during the receive phase all elements are read out individually and no receive apodization is applied. As a next step, it might be considered to use an apodization function also in receive when calculating the Doppler power. In that way, US pressure variations as a result of a set transmit-apodization may be compensated and possibly a more accurate determination of the power center may be achieved.

The experiments were performed under various conditions to analyze the received Doppler signals and to evaluate the performance of the different DAA-methods. To this end, a limited number of experimental parameters as well as parameter settings of the DAA-methods have been changed. As a result, we found that when the heart is located at a fixed position throughout the measurements, bending of the array has a positive effect on the $\bar{\mathrm{SNR}}$, since multiple elements are directed toward the heart. However, bending the array also has the disadvantage that identification of the most suitable transmitting elements, or the location of the power center *C* for the window-based DAA-methods, is more challenging under dynamic conditions. In addition, the velocities projected on the curved array are effectively increased relative to a flat array. Before implementing a DAA method in clinical practice, more work needs to be done to find the optimal parameter settings. In addition, the current experimental setup allows investigating the performance only for one-dimensional change of fHL. A setup that allows changing the fHL into multiple directions with varying velocities may provide more insight into the limitations of the proposed DAA-methods in more realistic conditions.

The developed prototype has in total 37 elements. This allows for measuring the fHR in a measurement volume which is approximately five times larger compared to a clinically applied US transducer. In [34,35], new Doppler systems with a large measurement volume have already been described. However, the goal of these studies is to identify fetal movements within a small measurement volume. Moreover, the US transducer elements in these studies are not integrated into a single device and when positioned on the maternal abdomen still do not cover the required full measurement volume to guarantee continuous fHR measurements independent of fHL over a long recording time. In this research, the main purpose of the designed array was to evaluate the performance of the DAA-methods. The exact required measurement volume, and hence, the exact size and shape of the flexible array, has to be further investigated. To be able to measure the fHR for all fHL within the uterus more elements will be needed, especially when applying such a flexible array on the abdomen on a mother with large BMI. Increasing the number of elements within the array makes it even more important to optimize the power consumption. It needs to be further elaborated on how this will affect the costs of such a system.

We investigated the performance of the Doppler signal quality in terms of the average $\bar{\mathrm{SNR}}$, of all Doppler signals. In general, for fHR monitoring a single Doppler signal with signal quality high enough to estimate the fHR would be sufficient. This suggests that the maximal SNR received may be a suitable measure to compare the different DAA-methods. However, combining the information acquired with multiple elements improves accuracy and robustness of fHR estimation [36]. Therefore, the authors assume that having an improved $\bar{\mathrm{SNR}}$ in the experiments will translate into a more robust and accurate determination of fHR in clinical practice. Clinical or preclinical measurements will be needed to validate this assumption.

In this research, the apodization is updated according to the Doppler power measured in the time window *w.* Consequently, the latest apodization function is always influenced by the location of the heart during that time window. If the movement direction of the fetal heart can be measured, it may be possible to predict the location of the heart during the next transmit-receive cycle, e.g., using a Kalman filtering approach [37], and one may consider this prediction when updating the apodization function.

It was shown that the curvature of the array has a large effect on the received Doppler power. The angle of incidence of the US waves may be different from the angle of reflection. In future research, the dynamic apodization method may be further improved by not only relying on the received Doppler power but also including information on the curvature of the transducer array.

Although this research focuses on the detection of the fHR, the proposed flexible transducer array has the potential of assessing the fetal activity, i.e., fetal movements, as well [35,38]. For that purpose, different filters need to be applied in the signal processing chain to be more sensitive to the slower tissue movements. Further, the DAA-method has to be refined to allow elements not directed toward the fetal heart but for example toward fetal limbs to transmit US waves, making fetal movement detection possible.

## 5. Conclusions

In conclusion, this research suggests that it is feasible to use a flexible transducer array to monitor the fetal heart rate for changing fetal heart location. In that way, the need for manual repositioning of the US transducer can be avoided, which significantly improves the clinical workflow. In addition, the proposed method to adapt the transmission power dynamically allows reducing the total acoustic dose transmitted to the fetus, which is an important clinical concern. Moreover, reduced power consumption of a Doppler monitoring method makes it more suitable to be used in an ambulatory setting.

## Figures and Tables

**Figure 1 sensors-19-01195-f001:**
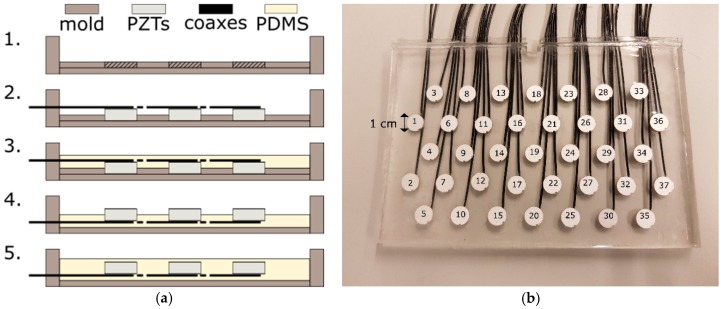
(**a**) Process of making a flexible transducer array. (1) Polymethylmethacrylat (PMMA) mold with plate for inserting (2) the piezotransducers. (3) Casting Polydimethylsiloxan (PDMS) on the backside of the transducers. (4) Flipping the first cast to mold a second layer of PDMS on the front side of the transducers (5) to create an air-free layer above the transducer elements. The processing steps are described in detail in Section 2.1; (**b**) Photograph of the flexible transducer array. For reference, the element number is shown on top of each transducer element.

**Figure 2 sensors-19-01195-f002:**
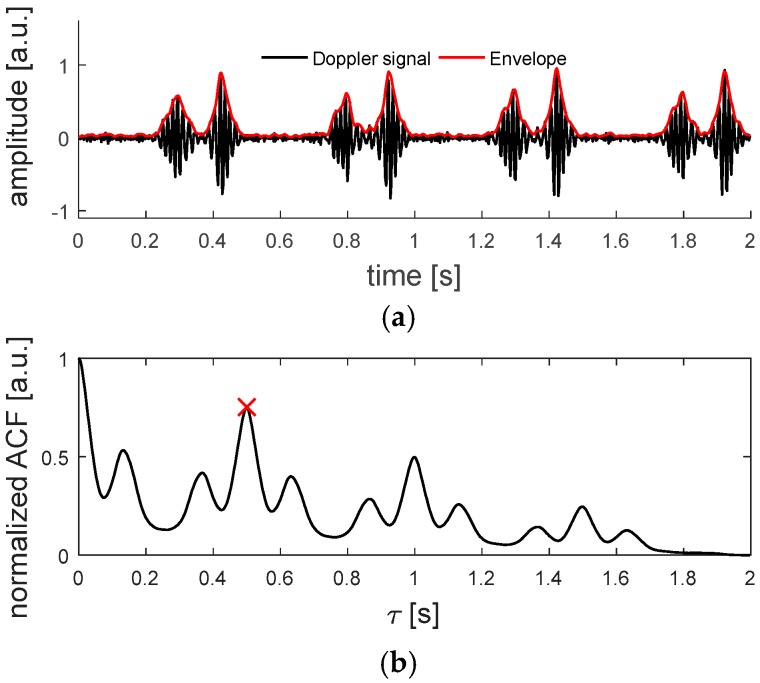
(**a**) Doppler signal acquired with a single element of the flexible transducer array, measured on an in-vitro beating fetal heart setup. In (**b**), the autocorrelation function is depicted. The maximum found at τ = 0.5 s corresponds to a heart rate of 120 bpm.

**Figure 3 sensors-19-01195-f003:**
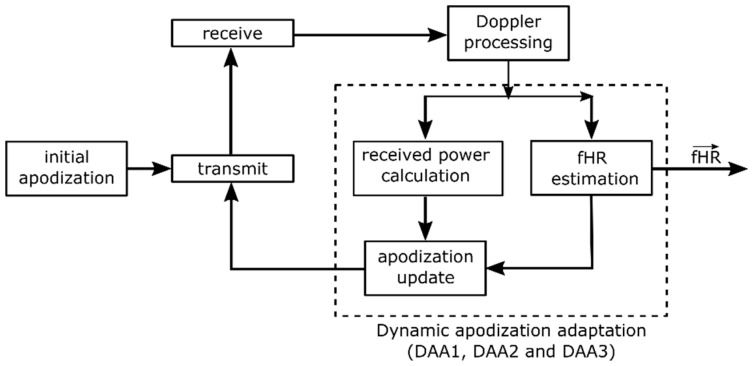
Block diagram of the dynamic apodization adaptation (DAA1, DAA2 and DAA3) process. An initial apodization is used in a first transmit-receive sequence. From the resulting Doppler signals, the power and fHR are estimated and used to update the apodization function, which modulates the transmission pulse of the next transmit-receive sequence.

**Figure 4 sensors-19-01195-f004:**
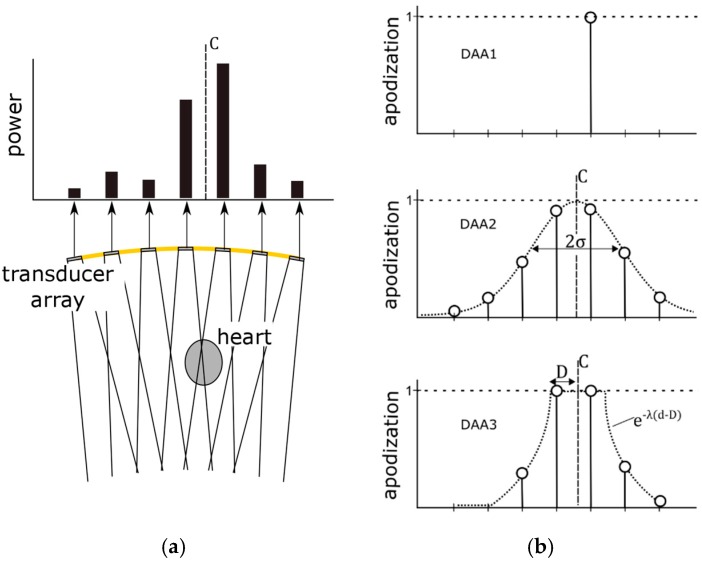
Illustration of the working principle of the dynamic apodization adaptation: (**a**) Received Doppler power measured in the individual transducer elements of the flexible transducer array. (**b**) For the DAA1-method, the apodization of a single element, which received the highest Doppler power, is set to 1. For the window-based apodization methods DAA2 and DAA3, the power center *C* defines the center of the window-based methods and the apodization of the elements is set accordingly.

**Figure 5 sensors-19-01195-f005:**
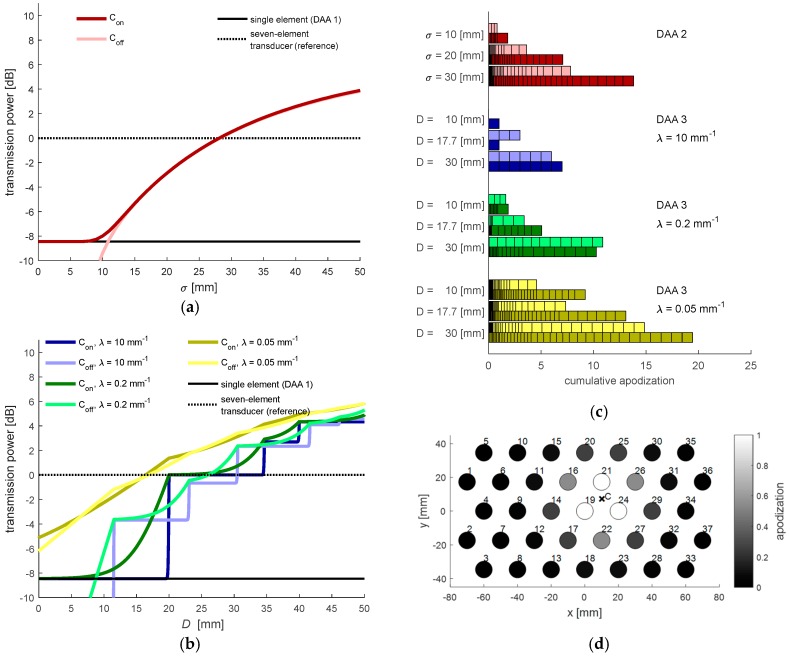
(**a**) Influence of the parameter *σ* (DAA2) on the total transmission power. The dark- and light-colored lines indicate the total transmission power depending on whether the power center *C* coincides (*C*_on_) with an element position or if it lies in between elements (*C*_off_). As reference, the dashed line at 0 dB indicates the total transmission power of a seven-element transducer used in clinical practice and the solid line is the transmission power when only a single element is used; (**b**) Influence of the parameter *D* (DAA3), for three selected decay parameters λ; (**c**) Cumulative apodization, the added apodization values set in all transducer elements, for selected parameter settings. Each block in the stacked horizontal bar plots corresponds to the apodization value of a single transducer element. The light and dark colored blocks correspond to the set apodization depending on whether the power center *C* coincides (*C*_on_) with an element position or if it lies in between elements (*C*_off_); (**d**) 2-dimensional representation of set apodization function with specific parameters (DAA3, *D* = 17.7 mm, λ = 0.05 mm^−1^).

**Figure 6 sensors-19-01195-f006:**
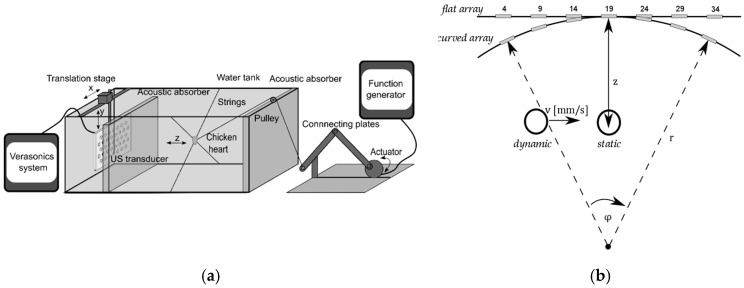
(**a**) Schematic of the fetal heart in-vitro setup. A function generator drives an actuator that is connected via a string to a chicken heart. In this way, the chicken heart is moved along the *z*-direction in a beat-like fashion. By translating the flexible transducer array through the water tank, a changing fetal heart location can be mimicked. (**b**) Illustration of the transducer array geometry and heart position during experiments.

**Figure 7 sensors-19-01195-f007:**
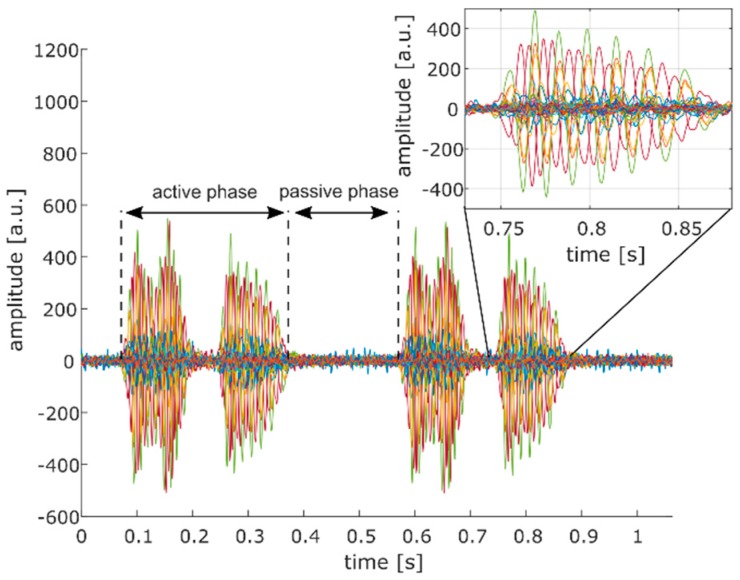
Definition of active and passive phase of the Doppler signal for the calculation of the SNR. Each colored line (37 in total) represents the Doppler signal of a single transducer element. The two events visible in the Doppler signal during the active phase correspond to forward and backward motion of the heart relative to the transducer. In the inset of the figure, strong phase variations of the different Doppler signals are visible.

**Figure 8 sensors-19-01195-f008:**
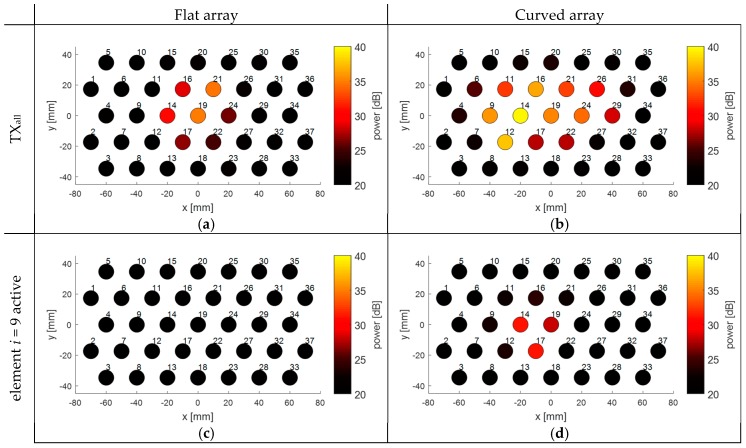
Effect of the curvature of the flexible transducer array on the received Doppler power in the individual elements. The number next to the transducer element indicates the element index *i*. The measurements were performed using an *in-vitro* beating heart setup. The heart was centered in front of the center of the array. (**a**) Received Doppler power when transmitting with all elements using a flat array; (**b**) Received Doppler power when transmitting with all elements using a curved array; (**c**) Received Doppler power when transmitting with element *i* = 9 only, using a flat array; (**d**) Received Doppler power when transmitting with element *i* = 9 only, using a curved array.

**Figure 9 sensors-19-01195-f009:**
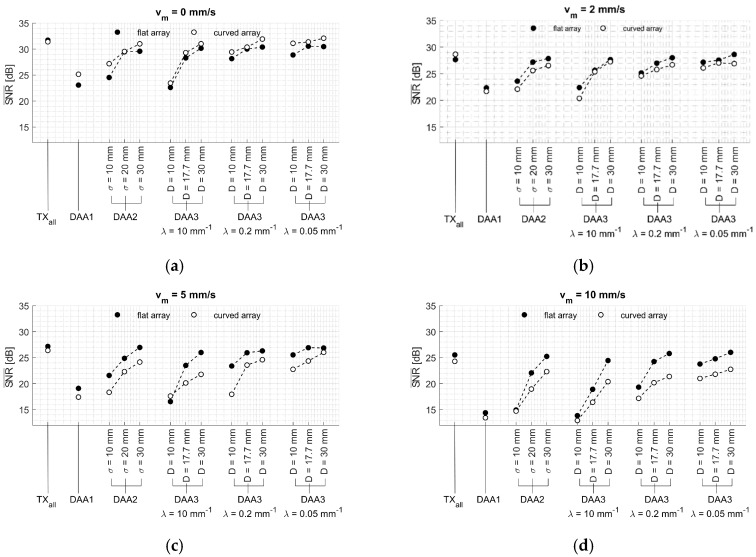
$\bar{\mathrm{SNR}}$ measured during the static (**a**) and dynamic experiments (**b**–**d**), with velocities *v_m_* = 2 mm/s, *v_m_* = 5 mm/s and *v_m_* = 10 mm/s respectively. Both the results using a flat (black) and a curved (white) array are depicted. On the horizontal axis, the different apodization function and, if applicable, parameter settings are indicated.

**Figure 10 sensors-19-01195-f010:**
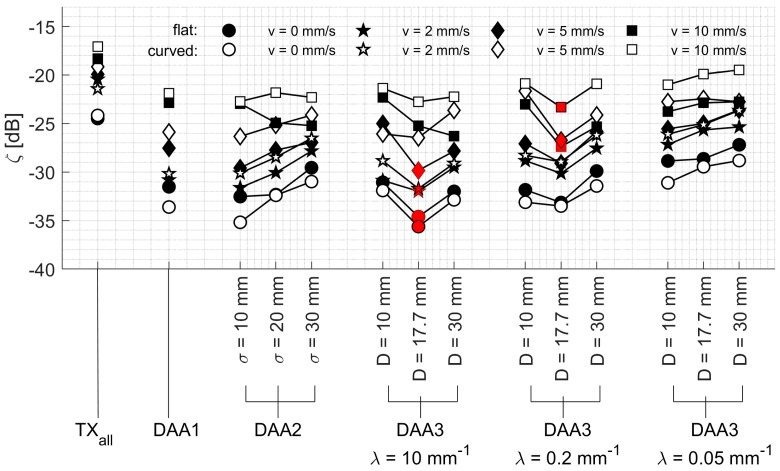
Cost function (Equation (7)) relating the measured $\bar{\mathrm{SNR}}$ to the total transmission power. The black and white color corresponds to the flat and curved array, respectively, and the different symbols correspond to the different velocities with which the experiments were performed. For each experiment, the method which has the lowest cost function is highlighted in red.

**Figure 11 sensors-19-01195-f011:**
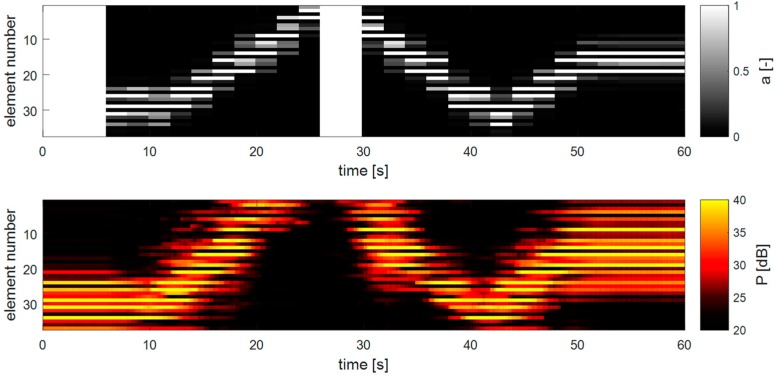
Dynamic apodization adaptation and received Doppler power during an in-vitro measurement. The flexible transducer array was curved during the experiment. After an initial time of 6 s, the DAA3 method with width *D* = 17.7 mm and λ = 0.2 mm^−1^ was used.

**Figure 12 sensors-19-01195-f012:**
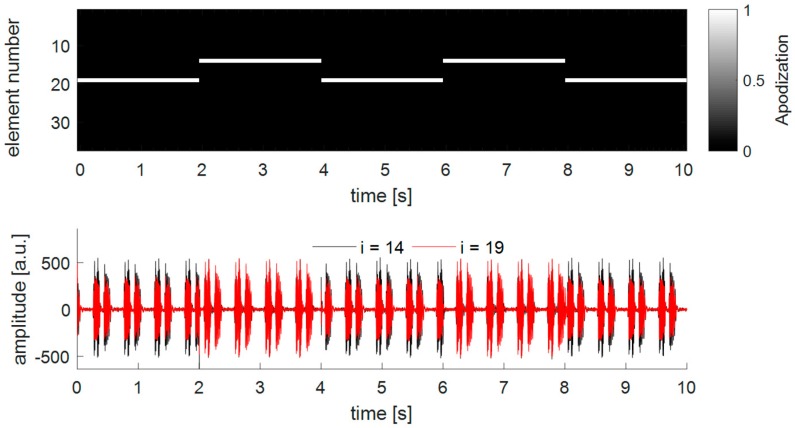
Illustration of a special behavior of the DAA1-method. In the experiment, the chicken heart is centrally located in front of a curved transducer array and beats at fHR = 120 bpm. During the experiments, the dynamic apodization adaptation method using a single element (DAA1) was used. The top graph shows the set apodization function over time and the bottom graph shows the Doppler signal received with element *i* = 14 and *i* = 19. For visualization, the time delay caused when updating the apodization function was removed from the apodization and Doppler signal graph.

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
