# Peer review of "Fetal Heart Rate Monitoring Implemented by Dynamic Adaptation of Transmission Power of a Flexible Ultrasound Transducer Array"

_sensors, 2019, doi:10.3390/s19051195_

Round 1

Reviewer 1 Report

This study presents a flexible Doppler Ultrasound (US) transducer array (applicable to arbitrary curvature of the abdomen) and 3 algorithms for optimal setting of the active transmitting elements so that to minimize the transmission power and to maximize the signal-to-noise ratio (SNR) in different locations of the fetal heart, as well as fetal heart movement in respect to the transducer. The methodological background, the experimental design and the obtained results are relevant and well described. The discussion and conclusions are supported by the results. The English style is very good so that the text is quite comprehensible. I have only a few minor revision remarks.  

1.       Abstract: The term "fetal heart (fHL)" to be replaced with "fetal heart location (fHL)". fHL should be also defined in the main text on its first use in Ln 50.

2.       Ln 42: "as they may directly reflect the functioning of autonomous regulation [9-13]" -> clarify the source – the mother's or the fetus's autonomous regulation?

3.       Ln 51: "recordings of the electrophysiological signal using abdominal surface electrodes" -> define the type of the electrophysiological signal under investigation. Are there any special requirements about the electrodes?

4.       Ln 53-54: Make sure that all abbreviations are defined on their first use, e.g. BMI, ECG are not defined. Check all over the text.

5.       Ln 67: " may puts" -> " may put"

6.       Ln 70-86: Both paragraphs are more relevant for section discussion, comparing old and new achievements, which the reader can judge after becoming familiar with the methodological details in section Methods. Instead, section Introduction should end with a short and compact definition of the aims of the study (presently missing). 

7.       Ln 87-88: This information is meaningless as soon as the structure of your paper is standard (Materials and Methods, Results, Discussion, Conclusions).

8.       Section 2.2. Ultrasound beam profile: The reader would become more convinced about the effectiveness of the new transducer array if its beam profile is theoretically depicted in a figure.

9.       Ln 197: "in-vitro beating fetal heart setup" -> should be described/depicted before showing the acquired Doppler signals (Figure 2).

10.    Figure 2: the x-axis of the ACF (bottom) to be displayed in units of seconds (not in samples) which seems more informative and corresponds to the Doppler signal (top)

11.    Ln 207-208: Clarify that the driving voltage of each acoustic element (and not all together) is adapted by the apodization function.

12.    Ln 226-229: Duplicated information with Ln 195-196 – seems unnecessary and confusing. Besides, the term "maximal velocity" should be explained so that it becomes clear how it is halved by DAA.

13.    Figure 3: The block "apodization update" should be clearly linked to the presented in the text apodization techniques. Therefore, it is suitable to show in the diagram the terms DAA1, DAA2, DAA3, which are further mathematically grounded.

14.    Ln 250, Ln 278: You speak about similar or improved SNR, however, the way for evaluation of this parameter should be earlier defined.

15.    Ln 281-283: “To distribute the transmission power among the transducer elements a window function can be used which defines the transmission power applied to the transducer elements.” -> improve the statement.

16.    Ln 286-288: “M’ are the elements which provide an fHR, which prevents that the apodization erroneously switches to elements which receive a high power only due to motion artefacts.” -> improve the statement.

17.    Ln 292: “Once C is determined” -> Equation (3) shows the calculation of C in case of flat transducer array. It is not clear how practically can be measured the coordinates (Xi, Yi) of the elements with a random position in a flexible array, considering that (Xi, Yi) should be substituted as arguments in (3) and (4).

18.    Ln 300-301: “The apodization for the elements surrounding this group, the apodization exponentially decays with rate λ.” looks more comprehensive in the form: “For the elements surrounding this group, the apodization exponentially decays with rate λ.”

19.    Figure 5 is first cited in section 2.5.2, but improperly placed in section 2.6.1. Similarly, Figure 7 is described in section 2.6.2, but misplaced in section 2.6.1.

20.    Section 2.5.3. (Re)-initialization should be part of experiments. It describes a specific setting, which is adopted just in your experiments, relevant to section 2.6.2. Experimental design.

21.    Figure 6 – make sure that the figure is not copied from [20], otherwise you have to ask for reproduction rights.

22.    Ln 563: "there parameters" -> correct the statement

23.    Ln 564: "This supports the clinically desired as-low-as-reasonably-achievable principle." -> clarify and give a reference to the principle.

24.    Section 4.4. Recommendation and conclusion" -> I advise the authors to split this long section into 4.4. Recommendations and 5. Conclusions

Ln 596-598: Improve the statement.

Author Response

Dear Reviewer,

We would like to thank you for your valuable feedback on our manuscript titled ‘Fetal heart rate monitoring implemented by dynamic adaptation of transmission power of a flexible ultrasound transducer array’.  Your feedback has been incorporated into the revised version and has led, in our opinion, to an improvement in clarity and readability of the manuscript.

Thank you and kind regards,

On behalf of all authors.

Paul Hamelmann

Reviewer 2 Report

The title as it is may be interpreted as "tested on fetal heart" whereas you did an (excellent) in vitro study. It might be considered to somehow hint this in the title, e.g. "In vitro demonstration of a flexible fetal heart rate monitoring ultrasound transducer array with dynamic transmission power adaptation" (this adds only 1 word). Note: just a suggestion.

If word count allows add "chicken" to "heart" in the abstractt.

Furthermore a very well written and elegant article!

Minor editorial: Suggest to also use (a) and (b) for figure 2 instead of referring to upper/lower halve.

Minor editorial: The placement of (a) near Vm=5mm/s and (b) near Vm=10mm/s is graphically giving a first impression to the eye that these values belong to each other, maybe some spatial re-arrangement could be tried.

Note these 2 suggestions are both just minor details. The article also stands well as it is.

Author Response

(The authors gave the same response as above.)
